

# Comparison of routine hematological indicators of liver and kidney function, blood count and lipid profile in healthy people and stroke patients

Xiaofang Cui[1,2], Wei Wei[1,3], Xiao Qin[1,3], Fei Hou[1,3], Jin Zhu[1,2] and Weiyang Li[1,3]

[1] Jining Medical University, Jining, Shandong, China
[2] Shandong Key Laboratory of Behavioral Medicine, School of Mental Health, Jining Medical University, Jining, Shandong, China
[3] Collaborative Innovation Center for Birth Defect Research and Transformation of Shandong Province, Jining Medical University, Jining, Shandong, China

## ABSTRACT

**Background and methods:** Stroke has become a major public health problem worldwide. In this article, we carried out statistical analysis, correlation analysis and principal component analysis (PCA) to evaluate the clinical value of routine hematological indicators in early diagnosis of ischemic stroke using R language.
**Results:** For the full blood count comparisons, stroke patients had obvious differences in the distribution width of red blood cells (RDW-CV), average distribution width of red blood cells (RDW-SD), mean hemoglobin concentrations, platelet large cell ratio, mean platelet volume and lymphocytes. Patients with ischemic stroke also exhibited different degrees of abnormalities in liver function test. With respect to renal function, stroke patients had obvious changes in uric acid and urea levels. Finally, when comparing the lipid profile, triglyceride concentrations were increased and high-density lipoprotein cholesterol concentrations were reduced in stroke patients. In addition, correlation analysis among these clinical indicators indicated that there were both common characteristics and differences between patients and health controls. Furthermore, the results of PCA indicated that these clinical indicators could distinguish patients from the healthy controls.
**Conclusion:** Conventional hematological clinical indicators, such as liver function, renal function, full blood count and lipid concentration profiles highly correlated with the occurrence of ischemic stroke. Therefore, the detection and analyzation of these clinical indicators are of great significance for the prediction of ischemic stroke.

# INTRODUCTION

Stroke is an acute cerebrovascular disease characterized by focal neurological deficits, which is due to ischemia in 60–80% of cases (*Doberstein et al., 2017*; *Global Burden of Disease Study Collaborators, 2015*). Stroke is common, with an incidence that increases yearly, and carries a high disability and mortality rate. Nowadays, stroke has become a

Corresponding author
Weiyang Li, 163.lwy@163.com

major public health problem all over the world (*Doberstein et al., 2017*; *Feigin et al., 2015*; *Global Burden of Disease Study Collaborators, 2015*). Thus, increasing our understanding of the clinical indicators for this disease is essential to improve the ability to predict stroke risk, as well as to reduce its incidence and associated disabilities. It is known that independent risk factors for stroke include age, sex, body mass index, positive family history, cigarette smoking, alcohol consumption, hyperlipidemia, hypertension, diabetes and heart disease (*Braillon, 2018*; *Guzik & Bushnell, 2017*; *Kim & Kim, 2018*; *Lan et al., 2018*). These factors may all cause changes in physiological functions thereby leading to abnormalities in renal or hepatic function, or blood count or lipid profile. Recent advances have been made in identifying biomarkers of acute ischemic stroke (*Arsava et al., 2011*; *Chatzikonstantinou et al., 2013*; *Maier et al., 2013*), but only a few systematic studies have been undertaken on the relationship between test indicators, such as liver function, renal function or full blood count and acute ischemic stroke (*Turcato et al., 2017a*, *2017b*). These bloods tests are routinely performed on hospital admission, and are minimally invasive and of low cost. In this study, we systematically evaluated the results of routine clinical blood tests to determine if there were obvious differences between patients with acute ischemic stroke and healthy individuals. Identification of easily performed clinical indicators that predict stroke is of great significance for early diagnosis and management of stroke.

## MATERIALS AND METHODS

Forty-one patients with acute ischemic stroke who were admitted to the Department of Neurology at Jining People's Hospital from April to December 2018 were enrolled as the disease group, and 80 healthy subjects were included as controls. The study had been approved by the Ethics Review Committee of the Jining Medical University and written informed consent was obtained from all subjects. For the stroke group, the sole inclusion criterion was that the patient was diagnosed with a first ischemic stroke event; patients with stroke accompanied by severe complications were excluded. Healthy individuals without obvious diseases were included in the healthy control group. Patients and healthy individuals with incomplete clinical data were excluded. Next, we assessed (1) renal function, (2) liver function, (3) full blood count, and (4) lipid concentrations. Data were processed by R language software, and the differences between two groups were examined by Student's *t*-test. Correlation analyze were carried out according to the method of Pearson correlation coefficient. PCA was performed based on the R language software.

## RESULTS

### Comparison of full blood count

Obvious differences were observed between the two groups in the distribution width of red blood cells (RDW-CV), average distribution width of red blood cells (RDW-SD), average hemoglobin concentration, average platelet volume, large platelet ratio and lymphocyte count as shown in Fig. 1. Compared with healthy individuals, the distribution of red blood cells in stroke patients was elevated, with increased mean hemoglobin concentration. Furthermore, compared with healthy individuals, the mean platelet

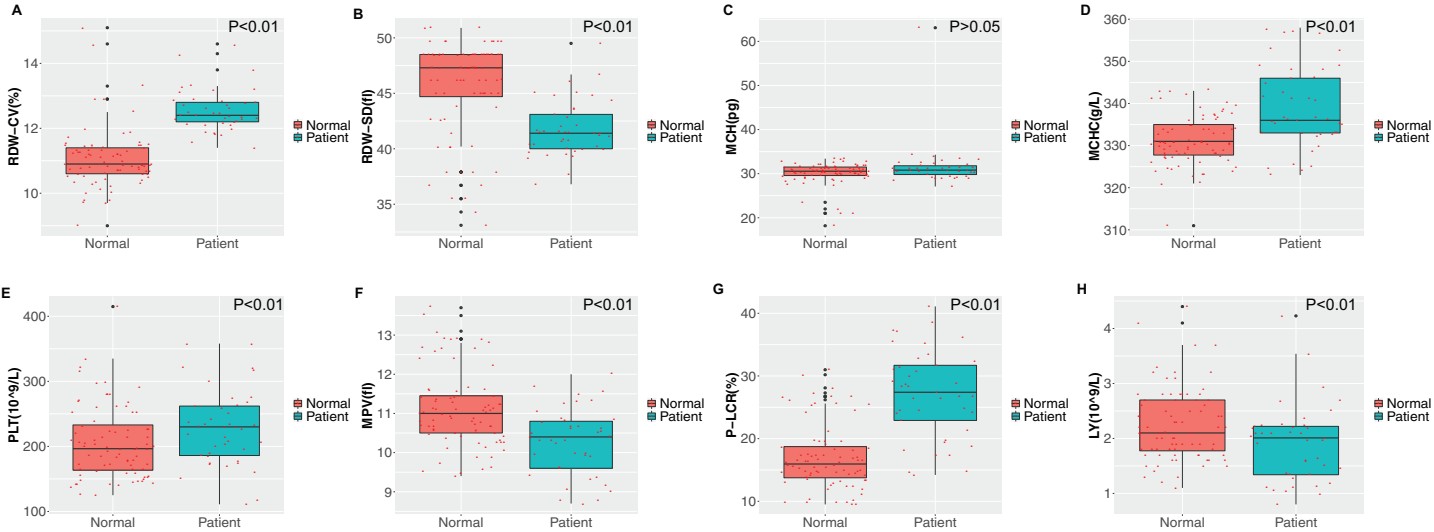

**Figure 1 Comparison of routine blood indicators.** (A) Red blood cell distribution width (RDW-CV); (B) mean red blood cell distribution width (RDW-SD); (C) mean corpuscular hemoglobin (MCH); (D) mean corpuscular hemoglobin concentration (MCHC); (E) platelet (PLT); (F) mean platelet volume (MPV); (G) platelet large cell ratio (P-LCR); (H) lymphocyte (LY). Green represents the patient group; Tangerine represents the normal group in the boxplot. The Student's $t$-test was used to calculate the $P$ value between groups.

volume in stroke patients was slightly lower ($P < 0.01$), while the large platelet ratio was elevated ($P < 0.01$). The lymphocyte concentration in stroke patients was lower compared to that in healthy individuals ($P < 0.01$).

## Comparison of liver function, lipid concentrations and renal function

We compared the liver function, renal function and lipid profile test results on admission between the two groups and the results are shown in Figs. 2 and 3. First, the liver function test results demonstrated that the albumin, albumin/globulin and total protein concentrations in stroke patients was lower than in healthy individuals ($P < 0.01$), while the globulin concentration was higher ($P > 0.05$). Further, stroke patients had lower indirect (unconjugated) and total bilirubin compared with the healthy group ($P < 0.01$). Second, comparison of the differences in blood lipid concentrations between the two groups revealed that the triglyceride concentration was higher and the high-density lipoprotein cholesterol concentration lower in stroke patients ($P < 0.01$). The differences in concentrations of the other lipids were not significant between the two groups ($P > 0.05$). Third, for evaluation of renal function, the uric acid concentration was notably reduced in the disease group, while the urea concentration was increased ($P < 0.01$).

## Correlation and principal component analyses of the clinical indicators between the two groups

Since multiple clinical indicators are associated with morbidity and mortality in patients with ischemic stroke, we correlated the various clinical indicators as shown in Fig. 4. Our findings indicate that there were correlations between the indicators in the disease group and in the healthy group, and that the correlation between the two groups was basically consistent with each other. However, the correlation of some indicators exhibited

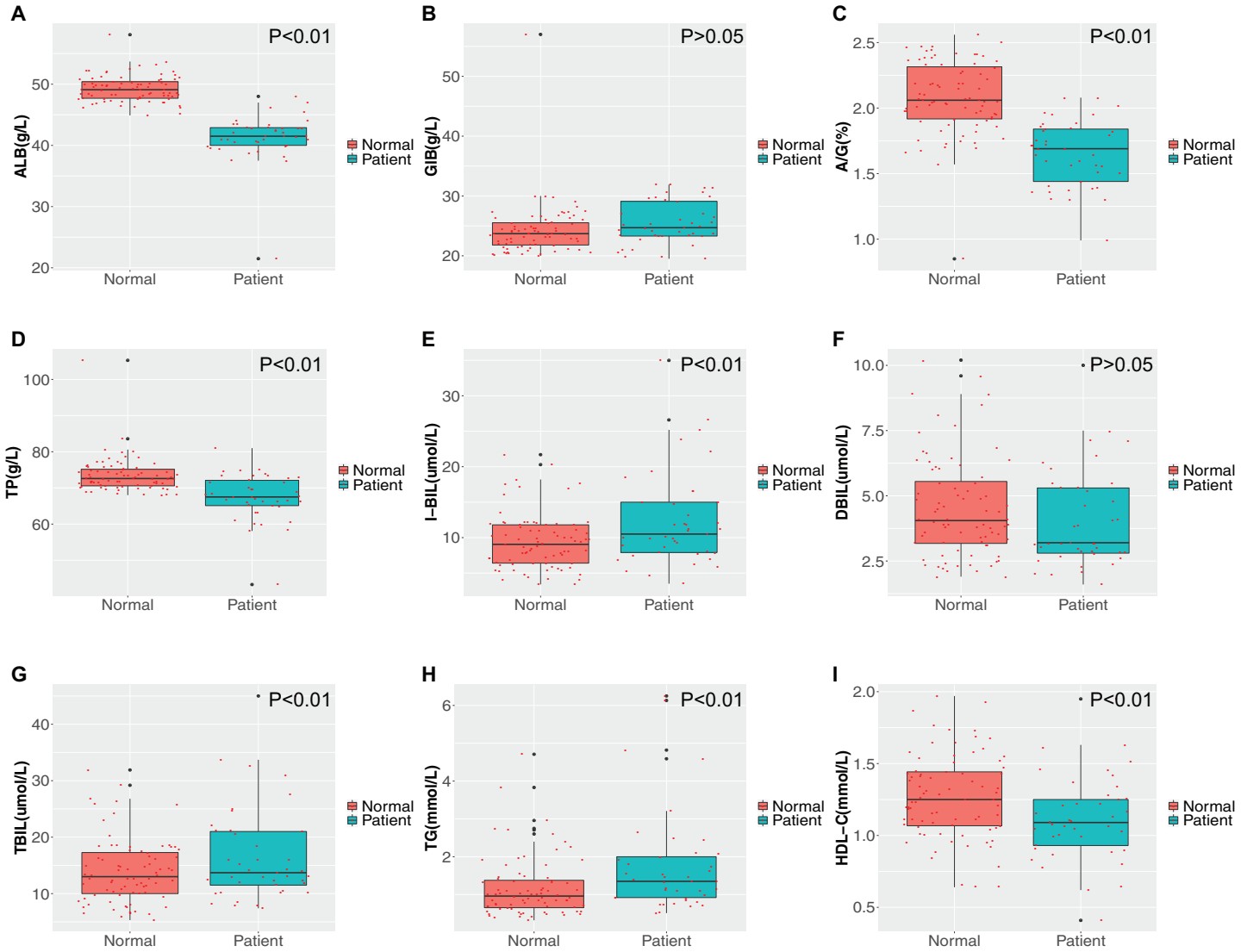

**Figure 2 Comparison of liver function and blood lipid indicators.** (A) Albumin (ALB); (B) globulin (GLB); (C) albumin/globulin (A/G); (D) total protein (TP); (E) indirect bilirubin (IBIL); (F) direct bilirubin (DBIL); (G) total bilirubin (TBIL); (H) triglyceride (TG); (I) high density lipoprotein cholesterol (HDL-C). Green represents the patient group; Tangerine represents the normal group in the boxplot. The Student's *t*-test was used to calculate the *P* value between groups.

remarkable differences between the two groups. For instance, the albumin/globulin ratio was negatively correlated with total protein in healthy individual, however this was not significant in the stroke group. Further, the high-density lipoprotein cholesterol concentration in the healthy group was negatively correlated with triglyceride concentration, however this was not observed in the patient group. In addition, there was an obvious negative association between mean cell hemoglobin and RDW-CV in the healthy group, which was not found in the patient group. In the healthy group, there was a positive correlation between mean cell hemoglobin concentration and mean cell hemoglobin, while in the patient group they were not significantly associated (Figs. 4A–4C). Due to the large number of clinical indicators and the significant differences

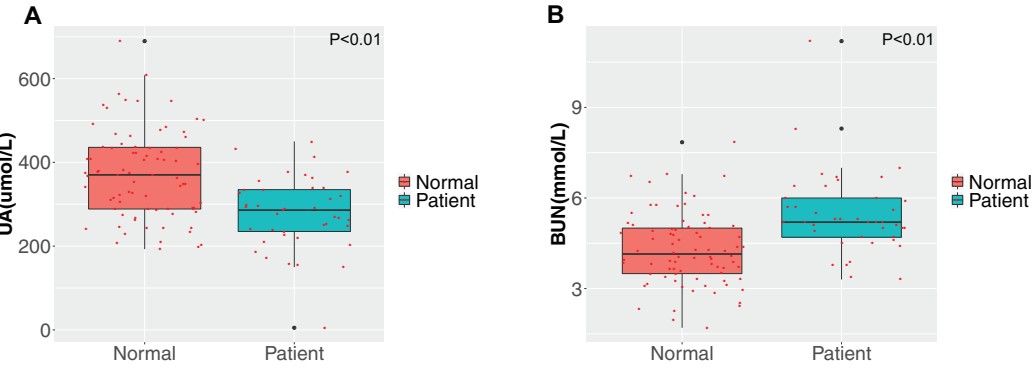

**Figure 3 Comparison of renal function.** (A) Uric acid (UA); (B) blood urea nitrogen (BUN). Green represents the patient group; Tangerine represents the normal group in the boxplot. The Student's *t*-test was used to calculate the *P* value between groups.

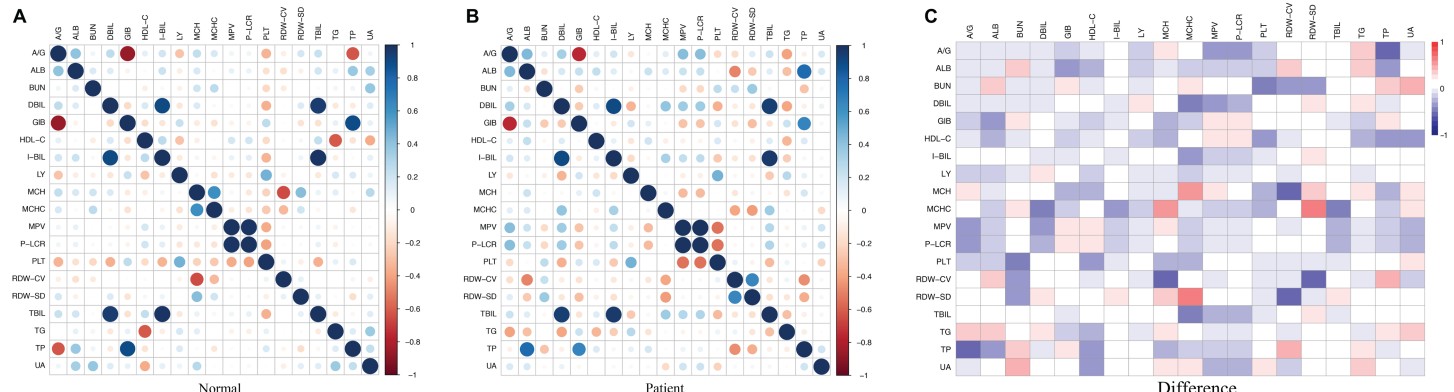

**Figure 4 Correlation of clinical indicators.** (A) Normal represents the correlation of clinical indicators in health group; the color represents the association strength. The Pearson Correlation Coefficient was used to calculate the correlation. (B) Patient represents the correlation of clinical indicators in stroke patients; the color represents the association strength. The Pearson Correlation Coefficient was used to calculate the correlation. (C) Difference represents the difference value of the correlation coefficient between Normal and Patient. The color represents difference value.

in multiple indicators between the healthy group and the stroke group, a comprehensive PCA was performed on the liver function tests, renal function tests, full blood count and lipid concentration profile. This clearly distinguished the healthy group from the disease group through differential indicators as shown in Fig. 5.

## DISCUSSION

Routine hematological testing in the clinical setting, that is important for diagnosis and management of diseases, includes evaluation of liver function, renal function, full blood count and lipid concentration profile. Although no systematic studies on the correlation between these routine clinical indicators and ischemic stroke have been performed either in China or abroad, a few reports have been published that evaluated the relationship between routine blood tests and stroke. Recent reports have been published on the relationship between the neutrophil/lymphocyte ratio (*Xue et al., 2017*; *Zhang et al., 2011*), red blood

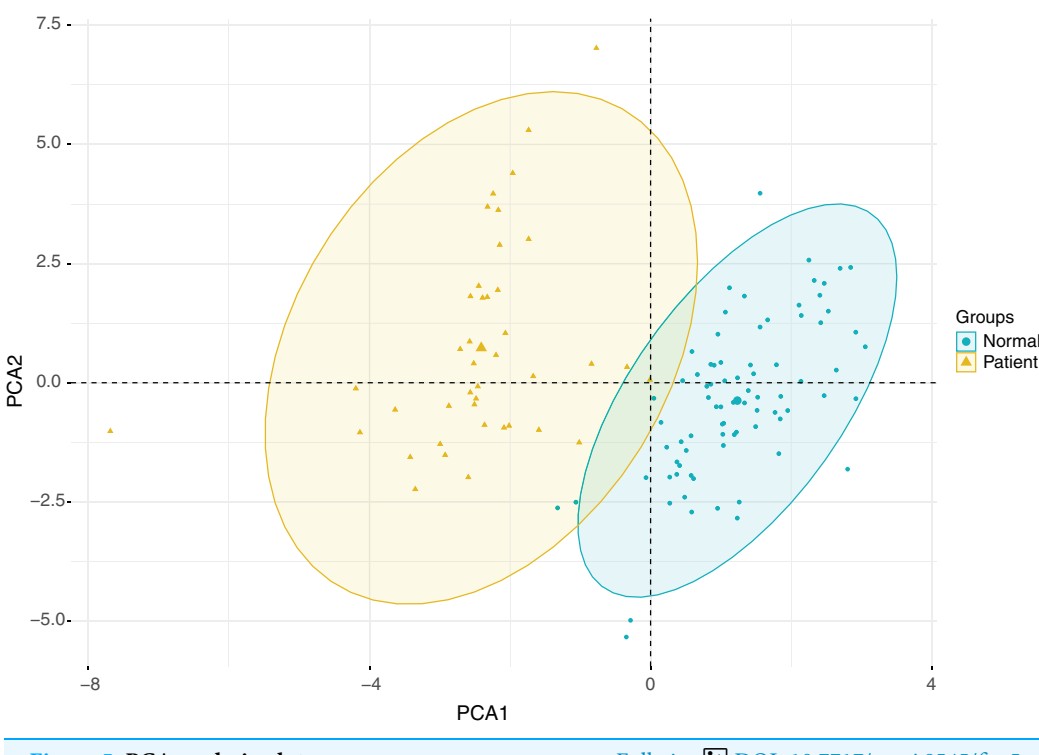

**Figure 5 PCA analysis plot.**

cell distribution width and stroke prognosis at home and abroad (*Kim et al., 2012a*; *Turcato et al., 2017b*). In 2017, *Turcato et al. (2017a)* found that determination of red blood cell distribution width could provide guidance on the risk of cardiovascular and cerebrovascular diseases. *Cao et al. (2011)* and *Zhang et al. (2012)* discovered that both the bilirubin concentration and hypertension were independently and negatively correlated with other risk factors. Since hypertension is one of the main risk factors for stroke, the occurrence of stroke may correlate with the abnormalities in liver function. Given that renal dysfunction often occurs during stroke episodes, a correlation between abnormal renal function and stroke may be possible. *Zhang et al. (2013*, *2012)* confirmed a strong correlation between elevated urea concentrations and poor prognosis in stroke patients. Moreover, many studies have reported a relationship between blood lipid concentrations and stroke (*Cui et al., 2012*; *Lewis & Segal, 2010*). Dyslipidemia is accepted as one of the risk factors for stroke because it can cause atherosclerosis, which is one of the pathological processes underlying stroke (*Kim et al., 2012b*). Therefore, the blood lipid concentration profile of stroke patients is usually abnormal.

In order to determine the differences in the above-mentioned clinical indicators between stroke patients and the healthy controls, we completed a separate analysis on more than 40 indicators that were evaluated in patients with ischemic stroke. Red blood cell width distribution, mean erythrocyte hemoglobin content, platelet count, mean platelet volume, platelet large cell ratio and lymphocyte concentration were among those parameters that were notably different in stroke patients when compared with healthy controls. Liver function test results demonstrated abnormalities in liver function protein

and bilirubin metabolism in stroke patients compared with healthy controls. We also observed that the serum uric acid concentration was decreased in stroke patients, while the urea concentration was remarkably elevated. This was consistent with previous findings (Zhang et al., 2013). In addition, the triglyceride and high-density lipoprotein cholesterol concentrations of stroke patients were significantly different from those of healthy controls. Therefore, measurement of these indicators may help predict or identify stroke. Through correlation studies, we explored the correlations between the indicators in both the stroke group and the healthy group, and significant differences were observed. The differences in correlations may be of great significance for predicting stroke. Subsequently, PCA was conducted on these indicators, and obvious differences were identified between the two groups. We found that these indicators could distinguish the stroke patients from the healthy controls. This comprehensive analysis of these indicators showed significant differences between the two groups, thereby presenting the ability to distinguish the two groups, and provide a theoretical basis for using these conventional indicators to diagnose stroke. In summary, early detection of changes in routine hematological indicators of liver and renal function, along with blood count and lipid concentrations in patients with ischemic stroke may have significant potential for early diagnosis, dynamic monitoring and evaluation of therapeutic effects.

### Funding
This work was supported by the Natural Science Foundation of Shandong Province (ZR2018PH018) and Jining Medical University (JY2017JS004). The funders had no role in study design, data collection and analysis, decision to publish, or preparation of the manuscript.

### Grant Disclosures
The following grant information was disclosed by the authors:
Natural Science Foundation of Shandong Province: ZR2018PH018.
Jining Medical University: JY2017JS004.

### Competing Interests
The authors declare that they have no competing interests.

### Author Contributions
- Xiaofang Cui conceived and designed the experiments, authored or reviewed drafts of the paper, and approved the final draft.
- Wei Wei analyzed the data, prepared figures and/or tables, and approved the final draft.
- Xiao Qin performed the experiments, prepared figures and/or tables, and approved the final draft.
- Fei Hou analyzed the data, prepared figures and/or tables, and approved the final draft.

- Jin Zhu analyzed the data, prepared figures and/or tables, and approved the final draft.
- Weiyang Li conceived and designed the experiments, authored or reviewed drafts of the paper, and approved the final draft.

## Human Ethics

The following information was supplied relating to ethical approvals (i.e., approving body and any reference numbers):

This study was approved by the Ethics Committee of Jining Medical University (2019-JC-011).

## Data Availability

Raw data are available in the Supplemental Files.

## Supplemental Information

Supplemental information for this article can be found online at http://dx.doi.org/10.7717/peerj.8545#supplemental-information.

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
