# Peer review of "Comparison of routine hematological indicators of liver and kidney function, blood count and lipid profile in healthy people and stroke patients"

_PeerJ, doi:10.7717/peerj.8545_

## Round 0.1 · original submission · Major Revisions

Please address all the critiques of both reviewers and amend your manuscript accordingly

Reviewer 1 ·

Basic reporting

The manuscript is well organized, however, the English language should be improved thoroughly. Some sentences are ambiguous, as listed below.

The introduction and background are clear and correctly referenced.

The structure of the manuscript conforms to PeerJ standards and discipline norm.

Figures are relevant, and appropriately labeled and described. All raw data are available.

The manuscript is self-contained and includes all relevant results.

Experimental design

The research is original, within Aims and Scope of the journal.

The research question is clearly defined, relevant and meaningful.

More details of the methods for statistical analysis should be described.

Validity of the findings

All underlying data have been provided and conclusions are well stated.

Additional comments

In the manuscript (#42702) entitled “Comparison of routine hematological indicators of liver and kidney function, blood count and lipid
profile in healthy people and stroke patients”, the authors systematically studied more than forty clinical indicators in 41 ischemic stroke patients with no other severe complications and 80 healthy people. Using statistical analysis, correlation analysis, and principal component analysis, the authors found abnormalities in conventional hematological clinical indicators in the stroke patients compared with the health people, such as red blood cell width distribution, platelet large cell ratio, and albumin concentration. Some of these differences are significant, and the patients and the health people can be clearly distinguished with sophisticated data analysis. Therefore, the detection and analyzation of these low-cost routine clinical indicators has great significance for risk prediction, early diagnosis and management of ischemic stroke.

Moderate comments:

1. The authors used the word “significantly” a lot in describing the difference in clinical indicators between patients and healthy people, mainly based on the P value. However, we know it has drawbacks to present and interpret results using P values. For example, although they all have a P value smaller than 0.01, the difference in RDW-CV is so obvious, but in LY is not large (in my opinion) and the P values could be flipped between significant and non-significant with different sets of data of the same parameter, especially when the sampling size is not large enough. The authors may consider using more neutral words to describe their results.

2. The method for statistical data analysis has not been described clearly. There is no description of box plots in the legends for Figs. 1−3. What type of correlation did the authors use? For PCA, did you use raw data or standardized data?

3. The authors could add another panel in Fig. 4, showing the difference in correlation coefficients between patients and health people. The dominant pairs will stand out by themselves, and it will be easier for the authors to interpret and audiences to get the information.


Minor changes to make the description more clear

Font size of Axis Labels in Figs 1, 2 is too small to read

“TP” is not mentioned in Fig 2 Legend

Lines 33-34. Reorganize the sentences “To evaluate … the statistic analysis” to “In this paper, we carried out statistical analysis, correlation analysis and principal component analysis to evaluated the clinical value of routine hematological indicators in early diagnosis of ischemic stroke using R language”

Line 38. Reorganize the sentence “different degrees of liver function test abnormalities” to “different degrees of abnormalities in liver function test”

Lines 42-43. This sentence is ambiguous. Make it clear.

Line 47. Replace “detection” by “the detection and analyzation”

Line 58. Replace “predictive” by “predict”

Line 65. Replace “only few” by “only a few”

Line 83. Reorganize the sentence “calculated by correlation and principal component (PCA) analysis.” to “examined by correlation analysis and principal component analysis (PCA).”

Line 90. Replace “Fig 1” by “Figure 1”

Line 98. Replace “Firstly” by “First”

Line 135. Replace “in China or abroad” by “either in China or abroad”

Lines 163-165. The sentence “These results demonstrated … the occurrence of stroke” is ambiguous. Make it clear.

Reviewer 2 ·

Basic reporting

Xiaofang Cui et al. investigated whether routine hematological indicators can be used to predict ischemic stroke. The authors critically compared the conventional hematological parameters in ischemic patients and the healthy person. The authors successfully established that these clinical parameters can be used to predict ischemic stroke. Professional English was used throughout the article. The introduction properly described the background and the context of the work with appropriate reference. The structure of the article conforms to the PeerJ standard. The figures were relevant and well-labeled and described. Raw data were provided for the article.

Experimental design

It was clearly stated how the research fills the stated knowledge gap in the introduction section. The research questions are well-defined and meaningful. The methods described in the manuscript are sufficient to follow. I have the following questions regarding the selection of patients for ischemic stroke.
Could you please elaborate “For the stroke group, the sole inclusion criterion was that the patient was diagnosed with a first ischemic stroke event;” Are all the patients already suffered an ischemic stroke? Was there a patient who didn’t suffer ischemic stroke yet but are likely in the future? If so, what are the parameters used for diagnosis? What are the timelines for occurrences of stroke and the collection of data on hematological indicators?

Validity of the findings

Data was interpreted completely and supported the conclusion. The conclusions were well described and in accordance with the original research questions. The authors were successfully established that the conventional hematological clinical indicators can be used to predict ischemic stroke.

Additional comments

Minor comments:
- The written informed content is in Chinese. Please provide a version translated in English.
Typos:
- Line 34: “stroke. R language was use to carried out the statistic analysis” - - > “stroke, the R language was used to carry out the statistical analysis”
- Line 37: “concentrations and platelet large cell ratio” - - > “concentrations, platelet large cell ratio”

---

## Round 0.2 · accepted · Accept

Both reviewers agreed that you have adequately addressed all their concerns in the revision, and the language has been improved. Therefore I am happy to accept your manuscript.

Reviewer 1 ·

Basic reporting

no comment

Experimental design

no comment

Validity of the findings

no comment

Additional comments

Increase the font size of Figure 4

Reviewer 2 ·

Basic reporting

The authors satisfactorily answered all the comments and provided the requested information.

Experimental design

No comment

Validity of the findings

No comment